# Severe Infantile Axonal Neuropathy with Respiratory Failure Caused by Novel Mutation in X-Linked *LAS1L* Gene

**DOI:** 10.3390/genes13050725

**Published:** 2022-04-21

**Authors:** Agnieszka Stembalska, Małgorzata Rydzanicz, Wojciech Walas, Piotr Gasperowicz, Agnieszka Pollak, Victor Murcia Pienkowski, Mateusz Biela, Magdalena Klaniewska, Zuzanna Gamrot, Ewa Gronska, Rafal Ploski, Robert Smigiel

**Affiliations:** 1Department of Genetics, Medical University, 50-368 Wroclaw, Poland; agnieszka.stembalska@umw.edu.pl; 2Department of Medical Genetics, Medical University of Warsaw, 02-106 Warsaw, Poland; piotr.gasperowicz@gmail.com (P.G.); poli25@wp.pl (A.P.); victorabel.murcia@gmail.com (V.M.P.); rploski@wp.pl (R.P.); 3Paediatric and Neonatal Intensive Care Unit, University Hospital in Opole, 45-401 Opole, Poland; wojciechwalas@wp.pl; 4MMG, Marseille Medical Genetics U1251, Aix Marseille University, 13385 Marseille, France; 5Department of Family and Paediatric Nursing, Medical University, 50-996 Wroclaw, Poland; mateuszbiela14@gmail.com (M.B.); magdazdzie@gmail.com (M.K.); 6Care and Therapy Unit for Mechanically Ventilated Children and Young People, 41-506 Chorzow, Poland; z.gamrot@gmail.com (Z.G.); ewagro@interia.pl (E.G.)

**Keywords:** infantile hypotonia, respiratory failure syndrome, SMARD, *LAS1L* gene

## Abstract

*LAS1L* encodes a nucleolar ribosomal biogenesis protein and is also a component of the Five Friends of Methylated CHTOP (5FMC) complex. Mutations in the *LAS1L* gene can be associated with Wilson–Turner syndrome (WTS) and, much more rarely, severe infantile hypotonia with respiratory failure. Here, we present an eighteen-month old boy with a phenotype of spinal muscular atrophy with respiratory distress (SMARD). By applying WES, we identified a novel hemizygous synonymous variant in the *LAS1L* gene inherited from an unaffected mother (c.846G > C, p.Thr282=). We suggest that the identified variant impairs the RNA splicing process. Furthermore, we proved the absence of any coding regions by qPCR and sequencing cDNA using amplicon deep sequencing and Sanger sequencing methods. According to the SMARD phenotype, severe breathing problems causing respiratory insufficiency, hypotonia, and feeding difficulties were observed in our patient from the first days of life. Remarkably, our case is the second described patient with a SMARD-like phenotype due to a mutation in the *LAS1L* gene and the first with a variant impacting splicing.

## 1. Introduction

Mutations in the gene *LAS1L*, which encodes a nucleolar ribosomal biogenesis protein—a component of the Five Friends of Methylated CHTOP (5FMC) complex—can result in Wilson–Turner syndrome (WTS) [1]. This is an X-linked neurological disorder characterized by severe intellectual disability, dysmorphic facial features, hypogonadism, short stature, and truncal obesity in affected males. Females with the mutation have no symptoms or have a milder phenotype than the affected males (OMIM:309585). There is also a less known but critical association between the mutations in the *LAS1L* gene and spinal muscular atrophy with respiratory distress (SMARD), known as severe infantile axonal neuropathy with respiratory failure (OMIM: 604320) [2,3]. SMARD is characterized by distal and proximal muscle weakness and diaphragmatic palsy that leads to respiratory distress and, without medical intervention, ultimately death before the age of 2 [4]. About one-third of cases of SMARD in the population are connected with the pathogenic variant of the *IGHMBP2* gene, which is inherited in an autosomal recessive manner. Apart from individual reports on the association of SMARD with such genes as *LAS1L* and *UBE1*, the etiology of the rest of the cases remains unknown [3,5,6].

We report an eighteen-month-old boy *with a phenotype of SMARD due to a novel hemizygotic variant in the LAS1L* gene. The variant was inherited in an X-linked recessive manner from the mother of the child.

## 2. Case Report

The proband is the first (GI, PI) child of unrelated, healthy parents. He was born at term by vaginal delivery with a birth weight of 3450 g. The Apgar score was 2/6/7/8 points. Prenatal history was not available, as the pregnancy was not controlled by gynecologists. No congenital defects or dysmorphic features after birth were noticed. However, breathing problems, muscular hypotonia, and feeding difficulties were observed from the first days of life. With time, respiratory insufficiency developed. The child initially required non-invasive respiratory support because of tachypnea, which resulted in intubation and mechanical ventilation. A paralyzed diaphragm on both halves of the diaphragm was suspected in a chest x-ray examination. Any attempt at extubating the child failed because of apnoea with a decrease in heart rate. Finally, tracheotomy was performed to continue mechanical ventilation. Feeding was implemented by using a gastric tube.

No abnormalities were found in echocardiography, ultrasonography of abdomen, and ophthalmological examination. During neurological examination of muscular hypotonia in the main axis of body, as well as in the limbs, wrist-drop was observed. Electroencephalography testing did not document evident pathological activity, but anti-epileptic drugs were applied because of clinical seizures. The results from a brain MRI (performed twice) showed transverse sinuses thromboembolism and a small cerebellum. Moreover, hypothyroidism was diagnosed and thyroid hormone supplementation was introduced. The childcare is currently palliative.

The following metabolic, hematologic, and genetic tests were performed with normal results: MS/MS, GC/MS, aminoacides profile in the blood and cerebro-spinal fluid, ammonia level, VIII factor activity, von Willebrand factor activity of the level of S-protein, Leiden mutation analysis, karyotype, MS-MLPA test for PWS, and SMN1 gene mutation analysis.

The current condition of the 18-month-old boy is severe but stable. He presents psychomotor development delay with reduced muscular tension in the head–torso axis and increased tension in the limbs, as well as sporadic eye contact. The proband still requires assisted ventilation in S/T AVAPS mode by tracheostomy (trouble-free ventilation). Feeding is carried out using gastrostomy. The boy presents severe skin lesions resulting from the atopic dermatitis accompanied by pruritus. No seizure episodes have been observed over the past year (pharmacological treatment with clonazepam and phenobarbital was still included). Due to the lack of complete recanalization of the cerebral vessels, he continues anticoagulation therapy.

## 3. Genetic Testing

### 3.1. Whole Exome Sequencing (WES)

Molecular diagnostics were performed using whole exome sequencing (WES). Venous blood samples were collected from the proband and his parents. WES was prepared using proband’s DNA sample with SureSelectXT Human kit All Exon v7 (Agilent, Agilent Technologies, Santa Clara, CA) and paired-end sequenced (2 × 100 bp) on HiSeq 1500 (Illumina, San Diego, CA, USA). The mean depth of coverage in the sequenced sample was 98×, with 98% of target covered ≥ 10×, and 94.7%, which covered ≥ 20×. A bioinformatics analysis of the raw WES data and variants prioritization were performed as previously described [7]. In brief, reads were aligned to the GRCh38 (hg38) reference genome with the Burrows–Wheeler Alignment Tool (http://bio-bwa.sourceforge.net/, accessed on 1 March 2022), and processed further by the Picard (http://broadinstitute.github.io/picard/, accessed on 1 March 2022) and Genome Analysis Toolkit (https://software.broadinstitute.org/gatk/ accessed on 1 March 2022). The identified variants were annotated with functional information, frequency in population (including gnomAD (http://gnomad.broadinstitute.org/ accessed on 1 March 2022), and an in-house database of > 5000 Polish exomes), and the known association with clinical phenotypes, which was based on both the ClinVar (https://www.ncbi.nlm.nih.gov/clinvar/, accessed on 1 March 2022) and HGMD (http://www.hgmd.cf.ac.uk, accessed on 1 March 2022) databases. An in-silico pathogenicity prediction was performed based on the pathogenicity and conservation scores available from the Varsome database (https://varsome.com/, accessed on 1 March 2022), including BayesDel addAF, BayesDel noAF, EIGEN, EIGEN PC, FATHMM-MKL, FATHMM-XF, LRT, M-CAP, MutPred, MutationTaster, PROVEAN, PrimateAI, SIFT, DEOGEN2, FATHMM, LIST-S2, MVP, Mutation assessor, and SIFT4G. Additionally, for the prediction of splicing defects, the following in silico tools were used: ADA [7], SPiCE [8], HSF [9], and Alamut v.2.11.0 (accessed on 1 March 2022). A prioritized variant was validated in the proband and his parents by amplicon deep sequencing (ADS), which was performed using the Nextera XT Kit (Illumina San Diego, CA, USA) and sequenced on HiSeq 1500 (Illumina, San Diego, CA, USA) as described above for WES.

### 3.2. Gene Expression (qPCR)

We performed a comparative qPCR procedure to evaluate *LAS1L* gene expression. Three samples were tested: proband, proband’s mother, and an unrelated healthy control. RNA was isolated from fibroblast pellets using the RNeasy Micro Kit (Qiagen, Hilden, Germany) with DNAse I digestion to remove any genomic DNA contamination. The isolation yielded high-quality RNA (RIN > 9), which then was used to obtain cDNA with the RevertAid First Strand cDNA Synthesis Kit (ThermoFisher Scientific, Waltham, MA, USA).

A comparative 2^−ΔΔCt^ qPCR was performed on a Quant Studio 12K Flex using SYBR Green PCR Master Mix reagents (ThermoFisher Scientific, Waltham, MA, USA). As an additional protection from genomic DNA amplification, primers were designed in accordance with the exon-exon junction principle. Exons 5–10 in the *LAS1L* transcript were amplified using the following primers: Forward: 5′- GGGAAGGGATAGAGGAAGAGG -3′; Reverse: 5′- CAAGTCCACGTTTTTGTGTGA -3′; amplicon size was 502 bp. We used two housekeeping genes: *ATP5F1* (Forward: 5′-GTCCAGGGGTATTACAGGCAA-3′, Reverse: 5′-TCAGGAATCAGCCCAAGACG-3′) and *GAPDH* (Forward: 5′-GCACAGTCAAGGCCGAGAAT-3′, Reverse: 5′-GCCTTCTCCATGGTGGTGAA-3′) as endogenous controls. All three samples combined with three genes (*LAS1L*, *ATP5F1,* and *GAPDH*) and the negative controls were tested in three replicates.

In addition to qPCR, we sequenced the cDNA of the patient, the patient’s mother, and a healthy control using NGS-based amplicon deep sequencing and Sanger sequencing methods to check the absence of any coding regions. NGS-based amplicon deep sequencing was performed using the same cDNA amplicons used for the qPCR analysis. Indexed libraries were constructed using Nextera XT Kit (Illumina, San Diego, CA, USA) and sequenced as described for WES analysis.

## 4. Results

By applying WES in the proband we identified a novel hemizygous synonymous variant in the *LAS1L* gene (hg38; chrX: g. 65529212C > G, NM_031206.7:c.846G > C, p.(Thr282=)). The presence of p.(Thr282=) was further confirmed by ADS in the proband and in his unaffected mother in heterozygotic form (Figure 1A). The p.(Thr282=) has 0 frequency in all tested databases, including gnomAD v3.1.2 and the in-house database of >5000 WES of Polish individuals. The c.846G > C is located in exon 6 out of 14 (residue 282 out of 735) of *LAS1L* and the c.846G position is highly conserved (PhastCons100way score 1.0). According to ACMG classification, the p.(Thr282=) variant was predicted as “likely pathogenic” (PP3 strong, PM2-moderate) [9]. Moreover. the p.(Thr282=) is suspected to impair the RNA splicing process. Thus, in silico tools for splicing defect predictions were employed to assess the impact of the p.(Thr282=) variant. All the tested predictors indicated a high probability of affecting the splicing at the 5′ donor site, including the ADA-score 0.9999, SPiCE-score 0.98964 [8], HSF-alteration of the WT donor site—which most probably will affect splicing [10]—and the Alamut v.2.11.0-predicted change of −31.1% (Figure 1B).

Sanger sequencing did not reveal any missing exons that could resulted from the affected splicing. However, qPCR based on fibroblasts material showed a 50% decrease of *LAS1L* gene expression in the proband when compared to his healthy control (Figure 2A). Furthermore, the results from cDNA deep amplicon sequencing identified in the proband, but not in the proband’s mother, a retained intron 6 (between exon 6 and 7). A detailed analysis revealed a dramatic drop in coverage starting at the genomic position chrX:65528876 (c.846 + 336C) from 357× to a single read at position g.X:65528875, which in fact showed a 336 bp partial intron 6 retention with a retained intron breakpoint at g.X:65528876. It is expected that the aberrant transcript results in a premature termination codon occurrence in the next 24 amino acids (Figure 2B). The codon stop could explain the decrease of the expression in the proband due to nonsense mediated decay (NMD).

## 5. Discussion

We present the patient with spinal muscular atrophy with respiratory distress caused by a novel hemizygotic variant of the *LAS1L* gene, which was inherited in an X-linked recessive manner from the mother of the boy. Our case is the second described patient with a SMARD-like phenotype due to a mutation in the *LAS1L* gene and the first with a variant impacting splicing. First, the association between a SMARD-like phenotype and a mutation in the *LAS1L* gene was observed by Butterfield et al. [3]. The authors described a boy with progressive respiratory insufficiency with diaphragm paralysis, muscular axial hypotonia with preserved antigravity limb movements, distal limb contractures (features in SMA-like or SMARD-like, with a fatally clinical outcome), and a de novo pathogenic mutation of c.1430G > A (p.S477N) found outside the functional domains in the *LAS1L* gene. This region is highly conserved in vertebrate species, and bioinformatic analyses showed the pathogenicity of the *LAS1L* mutation. This was confirmed using a zebrafish model by morpholinomediated knockdown of las1l. The mutation of Butterfield et al. was absent in the control population. The variant of *LAS1L* gene identified in our case is located on the intron–exon boundary and most probably affects mRNA splicing by producing in the proband (but not in carrier mother) splicing isoforms containing a partially retained intron between exon 6 and 7. We also confirmed that the alteration of splicing leading to the decreased dosage of *LAS1L* transcripts via (the most probably) NMD may be a molecular pathomechanism of the SMARD-like phenotype observed in our patient. Moreover, cDNA amplicon deep sequencing analysis revealed that the c.846G > C allele is not present in the maternal fibroblasts (Figure 2B), suggesting that the carrier mother might have skewed X chromosome inactivation with the X chromosome carrying the c.846G > C variant, which would be selectively inactivated. Thus, the c.846G > C is not expressed in the proband’s mother’s cells.

Our case, such as the case presented by Butterfield et al., suggests a common pathogenic mechanism, which is the disruption of ribosomal biogenesis [3]. The same negative influence for ribosomal maturation has mutations in the *IGHMBP2* gene, which are responsible for most known genetic cases of the SMARD phenotype [6,11]. Thus, we expect that the identified LAS1L p.(Thr282=) variant impairing the RNA splicing process may have a similar effect like the splice-site mutations described in IGHMBP. Since the presence of the full or partially retained intron is observed, it is expected to result in NMD or a truncated protein with an impaired or total lack of function [12,13].

Spinal muscular atrophy with respiratory distress (SMARD) is a group of rare disorders but is the second most common motor neuron disorder in infancy, and it is characterized by genetic heterogeneity (including inheritance) as well as clinical heterogeneity [4]. The phenotype of the more frequent type of SMARD (SMARD type 1) caused by mutations in the *IGHMBP2* gene mainly includes respiratory insufficiency due to diaphragmatic palsy, muscle weakness, and distal joint contractures [4,6,11]. There is no specific prenatal or early neonatal clinical signs of SMARD1. Intrauterine growth retardation, premature birth, hypotonia, weak cry and suckling, feeding problem, and foot malformation (fatty pads in the proximal phalanges) can be observed [4,6,14,15,16]. It is considered that early-onset respiratory involvement in children with a normal-shaped thorax caused by diaphragmatic palsy is the pathognomonic sign for SMARD1, unlike other neuromuscular disorders [4]. First, the symptoms can be shown on a chest X-ray, which is observed in the abnormal elevation of the diaphragm or hemidiaphragms [14,15,16]. Diaphragmatic palsy must be confirmed by chest ultrasound or diaphragmatic electromyography [6]. Respiratory failure, presented at birth (rare, not typical in SMARD) or usually between 6 weeks and 6 months of life, requires permanent ventilatory support, without which the patient with the severe form of the disease will die before 2 years of age [6,17]—when the first sign of respiratory failure appears [4,14,18].

Muscular involvement is the next most typical clinical finding observed in SMARD1. As a result of motor neuron death and neuromuscular atrophy, progressive muscular paralysis of the four limbs occurs [4,15]. Usually, the distal regions of the inferior limbs are the first, then the distal regions of the superior limbs, and, at the end, the proximal regions of all the limbs and trunk muscles are involved. Other clinical manifestations of SAMRD are progressive kyphoscoliosis, muscular atrophy in the feet and hands with adipose tissue accumulations in the phalanges (fatty pads) and with finger contractures, and congenital foot deformities [4,11,14]. Cognitive development in SMARD is believed to be normal [4,6,19].

There is no correlation between mutations in the *IGHMBP2* gene and phenotype. The differences in phenotype are observed in patients with the same genetic mutations.

The prognosis in SMARD is poor given that motor functions, as well as central (mimic muscle weakness, tongue fasciculations) and autonomic nervous system dysfunctions (bladder incontinence, urinary retention, excessive sweating, constipation, cardiac arrhythmia) are progressive with time, especially in the first two years of life [6]. After the second year of life a state stabilization sometimes occurs, and some functions can improve [10,19]. Eckart at al. proposed a clinical evaluation of the SMARD course in the form of the scoring system applied monthly in the first year of life and then yearly [18]. Earlier, the diagnostic criteria were proposed by Pitt et al. [18]. It is suggested, based on the age of onset of SMARD, that there are three forms of SMARD1: (1) an early-onset form, with symptoms before the 3rd month of age (very severe form); (2) a classical onset form; and (3) a late-onset form, with symptoms after the 12th month of age (slowly progressive form) [6,20]. Life expectancy with SMARD depends on the degree of respiratory failure and accompanied complications [19].

The phenotype of SMARD type 2 is caused by mutations in the *LAS1L* gene and is similar to SMARD1. Mutations in the *LAS1L* gene play a similar role in ribosomal biogenesis as the mutations in the *IGHMBP2* gene do, however, no direct interaction between these genes is demonstrated [3]. It is very difficult to say something about the genotype–phenotype correlation based on only two cases. Slight clinical differences between our patient and the one presented by Butterfield et al. are within the range of clinical heterogeneity.

Spinal muscular atrophy with respiratory distress is an untreatable disease. There are reports of attempts to use stem cells or gene therapy, but these methods are not currently available for patients [6]. Palliative care is needed in severe cases. The management is supportive, such as ventilatory support for respiratory failure and a gastrostomy tube placement for swallowing difficulties, aspirations, and poor caloric intake [4].

In every case of SMARD, genetic counselling must be performed. Given this, we informed the parents of our proband about all the aspects considered with this clinical situation. Although no other family members of the mother of our patient were examined, the mother received information about possible other female carriers of *LAS1L* gene mutation in her family. The mother was informed about pre-implantation and prenatal genetic diagnosis in the future because she has an increased risk of having another child with the same condition (50% of sons). She was informed about the risk of having a child carrier with the mutation (50% of daughters). To date, no symptomatic female carriers of SMARD have been reported [4,6].

## Figures and Tables

**Figure 1 genes-13-00725-f001:**
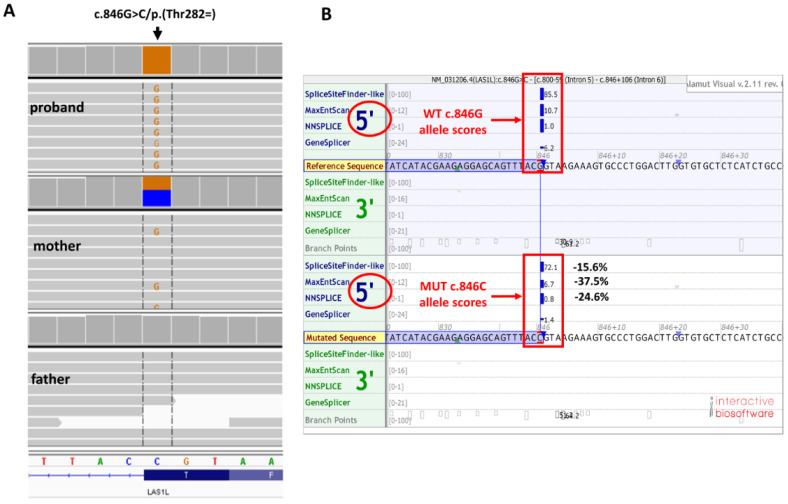
Results of genetic analysis. (**A**) Amplicon deep sequencing replication study results (Integrative Genomic Viewer screen shot) (**B**) Alamut in silico prediction of splicing effect of c.846G > C/p.(Thr282=) *LAS1L* variant. WT-wild-type allele, MUT–mutated allele. The scores next to vertical blue bars indicate predicted 5′ splice sites. Note the significant (≥10%) decrease of scores for c.846G > C variant in comparison with the wild-type sequence, which suggests splicing impairment; percentage of predicted change is given for the three main Alamut predicting tools.

**Figure 2 genes-13-00725-f002:**
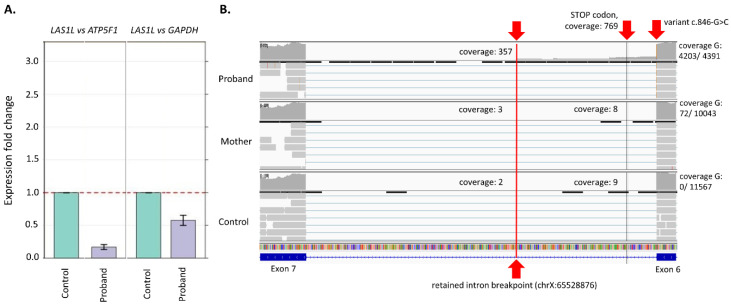
(**A**) Results from real-time PCR. A visible decrease of expression can be seen when the proband is compared to the control (**B**) Results from deep amplicon sequencing of cDNA. A partial retained intron is visible in the proband, while it is not present in the mother and the control. A partial retained intron breakpoint is marked with red line (g.X:65528876, from this position coverage drops significantly). Probably the decrease in the expression of *LAS1L* is caused by nonsense mediated decay due to the presence of a codon stop after 24 amino acids derived from the intronic sequence.

## Data Availability

The data presented in this study are available on request from the corresponding author.

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
