# Peer review of "Severe Infantile Axonal Neuropathy with Respiratory Failure Caused by Novel Mutation in X-Linked LAS1L Gene"

_genes, 2022, doi:10.3390/genes13050725_

Round 1
Reviewer 1 Report
In this manuscript the authors report on mutation in the LAS1L gene in a patient with a phenotype of SMARD. It is important to report cases of SMARD caused by LAS1L mutations, as most cases in the literature have been reported by loss of function mutations in the IGHMBP2 gene. The authors also include functional data from the patient fibroblasts to show that the variant results in a reduction in LAS1L expression by RT-PCR. It is predicted that the mutation results in a splicing change. I have some comments that I think should be addressed.
It would be helpful for the authors to indicate the frequency of this variant in databases such as gnomAD.
The alternating hypotonia and hypertonia seems unexpected. What do the authors attribute this finding to?
The MRI brain shows “transverse sinuses thromboembolism”. Was there more than one thromboembolism. Was diffusion weighted MRI performed to determine if the patient had a stroke? This would also modify the clinical presentation and would be important for the authors to address. What is the likely cause of the thromboembolism?
The results section should indicate that the expression data was from fibroblasts. Is the healthy unrelated control age matched?
Minor concerns:
-Part of figure 1A is cutoff “proband”
Author Response
Review 1
• In this manuscript the authors report on mutation in the LAS1L gene in a patient with a phenotype of SMARD. It is important to report cases of SMARD caused by LAS1L mutations, as most cases in the literature have been reported by loss of function mutations in the IGHMBP2 gene. The authors also include
functional data from the patient fibroblasts to show that the variant results in a reduction in LAS1L expression by RT-PCR. It is predicted that the mutation results in a splicing change. I have some comments that I think should be addressed.
Answer: Thank you so much for your detailed review. All comments and suggestions are extremely valuable to us. We hope that the corrections we made will be satisfying.
• It would be helpful for the authors to indicate the frequency of this variant in databases such as gnomAD.
Answer: As suggested we added the frequency of LAS1L p.(Thr282=) variant and changing the sentence in the Results section which is now as follows:
“The p.(Thr282=) has 0 frequency in all tested databases, including gnomAD v3.1.2 and in-house database of > 5000 WES of Polish individuals.”
• The alternating hypotonia and hypertonia seems unexpected. What do the authors attribute this finding to?
Answer: We corrected this sentence.
• The MRI brain shows “transverse sinuses thromboembolism”. Was there more than one thromboembolism. Was diffusion weighted MRI performed to determine if the patient had a stroke? This would also modify
the clinical presentation and would be important for the authors to address. What is the likely cause of the thromboembolism?
Answer: The true is that the transverse sinuses thromboembolism in MRI study of brain was suspected. In our opinion it could be caused by severe respiratory insufficiency. The is not central point of general clinical presentation of patient. We corrected this sentence.
• The results section should indicate that the expression data was from fibroblasts. Is the healthy unrelated control age matched?
Answer: We took sample from fibroblasts from proband, proband’s mother and unrelated healthy control. This information is included into the text (methods and results). “qPCR showed a 50% decrease of LAS1L gene expression in the proband comparing to his healthy male control” We corrected the sentence in the Results section: “qPCR based on fibroblasts material showed a 50% decrease of LAS1L gene expression in the proband comparing to his healthy male control”
Minor concerns:
• Part of figure 1A is cutoff “proband”
Answer: We are sorry for this neglect, we provided a new and complete Figure 1.
Reviewer 2 Report
Here the authors present a case of an infant with hypotonia and respiratory failure attributed to a non-coding variant in LAS1L thought to decrease transcription and affect splicing.
- It is probably an overstatement to suggest that “mutations in the LAS1L gene typically cause Wilson-Turner syndrome,” as mentioned in the abstract and also in the introduction. WTS had been only reported in two families with LAS1L variants. Mention of the association is fine, but the association is probably very rare for either phenotype, so could soften the language here.
- The phenotype description is not very extensive from a neurological standpoint, and is not very convincing for SMARD. There is no detailed description of the neurologic examination or the distribution of weakness, contracture, etc. Distal weakness including wrist-drop is a common feature in SMARD including the other reported SMARD case due to LAS1L mutation. There is no description of whether the patient has weak or paralyzed diaphragms (by chest x-ray or ultrasound for example). Involvement of the diaphragm out of proportion to the limb muscles is a hallmark of SMARD. Was there a muscle biopsy performed?
- Seizure-like episodes are mentioned but not well described. Workup included an EEG which did not capture the episodes. Is there an alternative explanation for these episodes? There is not nerve conduction or EMG results, which could confirm a severe neuropathy as cause for the hypotonia. Alternating hyper- and hypotonia is described which is not seen in SMARD cases. Do the authors have an explanation of the hypertonia? Is there an explanation of the hypothyroidism? Mutations in the X-linked gene MCT8 could be an explanation for both. Were there any variants seen on exome sequencing in this gene or others that could have overlapping phenotypes. There is no mention of how the presence of a cerebral venous sinus thrombosis could complicate the phenotype. From the author list, there does not appear to be a neurologist included, which would likely improve the phenotype description.
- Genetic workup prior to WES is not well described. What is the cytogenetic test and methylation test that were performed and what was the result?
- Metabolic workup is not well described or very extensive. Critically, thyroid testing results are not presented.
- Variant analysis is incomplete. There results mention that the c.846G>C variant seen in the patient was absent in all tested databases, but which databases were checked is not stated. There is no discussion of conservation at this position. While it is included in the results, the methods for the in-silico splice analysis are not described.
- The authors suggest the presence of a retained intron, but the data are incomplete on this. Figure 2 included some data from “deep amplicon sequencing,” but we are not given any information about how this was performed. The figure includes 7 nucleotides of intronic sequence with coverage of 963 reads, but does not include the full extent or position of the size of the intronic sequence included in the variant transcript. The authors suggest the intronic sequence leads to a premature stop but don’t show the full variant sequence. These data can be readily obtained from RNA-seq data that would show the exact inclusion of the intronic sequence and allow quantification of the variant transcript compared to the normal transcript. Can the authors explain why we don’t seem to see the proposed variant transcript in the carrier mother? Is there any evidence for a dosage effect from LAS1L? Is there literature among males with CNV including the LAS1L gene for example that could shed some light on this question.
Author Response
Review 2
- It is probably an overstatement to suggest that “mutations in the LAS1L gene typically cause Wilson-Turner syndrome,” as mentioned in the abstract and also in the introduction. WTS had been only reported in two families with LAS1L variants. Mention of the association is fine, but the association is probably very rare for either phenotype, so could soften the language here.
Answer: You have right, we changed the sentence to soften the conviction of the statement.: “ mutations in the LAS1L gene typically cause Wilson-Turner syndrome” for “can be associated with Wilson-Turner syndrome”.
- The phenotype description is not very extensive from a neurological standpoint, and is not very convincing for SMARD. There is no detailed description of the neurologic examination or the distribution of weakness, contracture, etc. Distal weakness including wrist-drop is a common feature in SMARD including the other reported SMARD case due to LAS1L mutation. There is no description of whether the patient has weak or paralyzed diaphragms (by chest x-ray or ultrasound for example). Involvement of the diaphragm out of proportion to the limb muscles is a hallmark of SMARD. Was there a muscle biopsy performed?
Answer: Muscle biopsy was not performed because of invasiveness of the test. We added some sentences in clinical description of patient. Thank you for your remarks.
- Seizure-like episodes are mentioned but not well described. Workup included an EEG which did not capture the episodes. Is there an alternative explanation for these episodes? There is not nerve conduction or EMG results, which could confirm a severe neuropathy as cause for the hypotonia. Alternating hyper- and hypotonia is described which is not seen in SMARD cases. Do the authors have an explanation of the hypertonia? Is there an explanation of the hypothyroidism? Mutations in the X-linked gene MCT8 could be an explanation for both. Were there any variants seen on exome sequencing in this gene or others that could have overlapping phenotypes. There is no mention of how the presence of a cerebral venous sinus thrombosis could complicate the phenotype. From the author list, there does not appear to be a neurologist included, which would likely improve the phenotype description.
Answer: Seizure-like episodes are mentioned but not well described. Workup included an EEG which did not capture the episodes. Is there an alternative explanation for these episodes? There is not nerve conduction or EMG results, which could confirm a severe neuropathy as cause for the hypotonia. Alternating hyper- and hypotonia is described which is not seen in SMARD cases. Do the authors have an explanation of the hypertonia? Is there an explanation of the hypothyroidism? Mutations in the X-linked gene MCT8 could be an explanation for both. Were there any variants seen on exome sequencing in this gene or others that could have overlapping phenotypes. There is no mention of how the presence of a cerebral venous sinus thrombosis could complicate the phenotype. From the author list, there does not appear to be a neurologist included, which would likely improve the phenotype description.
- Genetic workup prior to WES is not well described. What is the cytogenetic test and methylation test that were performed and what was the result?
Answer: Thank you for your this remarks. We corrected this sentence. cytogenetic test – karyotype, and methylation test – test against PWS (MS-MLPA)
- Metabolic workup is not well described or very extensive. Critically, thyroid testing results are not presented.
Answer: In our opinion the metabolic results and thyroid tests were not critical for our patient, so we removed the detail information from the main text.
- Variant analysis is incomplete. There results mention that the c.846G>C variant seen in the patient was absent in all tested databases, but which databases were checked is not stated. There is no discussion of conservation at this position. While it is included in the results, the methods for the in-silico splice analysis are not described.
Answer: as suggested, we provide more info about variant analysis. The improved parts of Methods and Results sections are now as follows:
Methods
“Bioinformatics analysis of raw WES data and variants prioritization were performed as previously described [7]. In brief, reads were aligned to the GRCh38 (hg38) reference genome with Burrows-Wheeler Alignment Tool (http://bio-bwa.sourceforge.net/), and processed further by Picard (http://broadinstitute.github.io/picard/) and Genome Analysis Toolkit (https://software.broadinstitute.org/gatk/). Iidentified variants were annotated with functional information, frequency in population (including gnomAD (http://gnomad.broadinstitute.org/), and an in-house database of > 5000 Polish exomes), and known association with clinical phenotypes, based on both ClinVar (https://www.ncbi.nlm.nih.gov/clinvar/) and HGMD (http://www.hgmd.cf.ac.uk) databases. In silico pathogenicity prediction was performed based on pathogenicity and conservation scores available from Varsome database, including BayesDel addAF, BayesDel noAF, EIGEN, EIGEN PC, FATHMM-MKL, FATHMM-XF, LRT, M-CAP, MutPred, MutationTaster, PROVEAN, PrimateAI, SIFT, DEOGEN2, FATHMM, LIST-S2, MVP, Mutation assessor, SIFT4G (https://varsome.com/). Additionally, for prediction of splicing defect the following in silico tools were used: ADA [7], SPiCE [8], HSF [9] and Alamut v.2.11.0. Prioritized variant was validated in the proband and his parents by amplicon deep sequencing (ADS) performed using Nextera XT Kit (Illumina) and sequenced on HiSeq 1500 (Illumina) as described above for WES.”
Results
“By applying WES, in the proband we identified a novel hemizygous synonymous variant in LAS1L gene (hg38; chrX: g. 65529212C>G, NM_031206.7:c.846G>C, p.(Thr282=)). The presence of p.(Thr282=) was further confirmed by ADS in the proband and in his unaffected mother in heterozygotic form (Figure 1A). The p.(Thr282=) has 0 frequency in all tested databases, including gnomAD v3.1.2 and in-house database of > 5000 WES of Polish individuals. The c.846G>C is located in exon 6 out of 14 (residue 282 out of 735) of LAS1L, and the c.846G position is highly conserved (PhastCons100way score 1.0). According to ACMG classification, the p.(Thr282=) variant was predicted as “likely pathogenic” (PP3 strong, PM2 - moderate) (PMID: 25741868). Moreover. the p.(Thr282=) is suspected to impair RNA splicing process. Thus, in silico tools for splicing defect prediction were employed to assess the impact of p.(Thr282=) variant. All tested predictors indicated high probability of affecting splicing at 5’ donor site, including ADA ‐score 0.9999, SPiCE‐score 0.98964 [8], HSF‐alteration of the WT donor site, most probably affecting splicing [9] and Alamut v.2.11.0‐predicted change −31.1% (https://www.interactive-biosoftware.com) (Figure 1B).”
- The authors suggest the presence of a retained intron, but the data are incomplete on this. Figure 2 included some data from “deep amplicon sequencing,” but we are not given any information about how this was performed.
Answer: A suggested we improved the description of cDNA deep amplicon sequencing. The modified part of Methods section is now as follow:
“NGS-based amplicon deep sequencing was performed using the same cDNA amplicons used for qPCR analysis. Indexed libraries were constructed using Nextera XT Kit (Illumina) and sequenced as described for WES analysis.”
- The figure includes 7 nucleotides of intronic sequence with coverage of 963 reads, but does not include the full extent or position of the size of the intronic sequence included in the variant transcript. The authors suggest the intronic sequence leads to a premature stop but don’t show the full variant sequence. These data can be readily obtained from RNA-seq data that would show the exact inclusion of the intronic sequence and allow quantification of the variant transcript compared to the normal transcript.
Answer: Figure 2 has been reworked according to the reviewer's suggestion. Intron sequence has been expanded so the stop codon is visible. Intron coverage values have been adjusted so they point at stop codon position.
- Can the authors explain why we don’t seem to see the proposed variant transcript in the carrier mother? Is there any evidence for a dosage effect from LAS1L? Is there literature among males with CNV including the LAS1L gene for example that could shed some light on this question.
Answer: Indeed, LAS1L cDNA amplicon deep sequencing analysis revealed that c.846G>C allele is not present in maternal fibroblasts (Figure 2B), suggesting that carrier mother might have skewed X chromosome inactivation with the X chromosome carrying the c.846G>C variant being selectively inactivated. Thus, the c.846G>C is not expressed in proband’s mother cells. An appropriate text was added to the Discussion section as follows:
“The variant of LAS1L gene identified in our case is located on intron-exon boundary and most probably affects mRNA splicing by producing in the proband (but not in carrier mother) splicing isoform containing retained intron between exon 6 and 7. Moreover, cDNA amplicon deep sequencing analysis revealed that c.846G>C allele is not present in maternal fibroblasts (Figure 2B), suggesting that carrier mother might have skewed X chromosome inactivation with the X chromosome carrying the c.846G>C variant being selectively inactivated. Thus, the c.846G>C is not expressed in proband’s mother cells.”
To the bast of our knowledge, there is no literature evidence for a dosage effect and CNVs in LAS1L gene. Thus, we performed a searching of different databases.
For a potential dosage effect of LAS1L we checked the ClinGen Dosage Sensitivity Map (https://www.ncbi.nlm.nih.gov/projects/dbvar/clingen/), however no evidence was reported.
For CNVs we searched the DECIPHER database (https://www.deciphergenomics.org/) for LAS1L CNVs, and among 47 patients with reported CNVs (both gains and losses) 11 were males (10 with duplications and one with deletion) presenting a wide spectrum of phenotypes, including among others intellectual disability, delayed speech and language development and hypotonia. Moreover, we also searched Database of Genomic Variants (DGV; http://dgv.tcag.ca/dgv/app/home) providing a catalogue of chromosomal structural variation (SV), including CNVs, indels and inversions, identified in healthy/clinically unaffected individuals, thus classified as non-pathogenic. Three SVs involving the complete sequence of LAS1L gene are reported, including two duplications and one inversion, however, the sex of individuals is not reported. Since CNVs of dosage-sensitive genes are not expected to be observed in healthy individuals, LAS1L seems not to be dosage-sensitive.
Reviewer 3 Report
In this article, the authors reported a clinical case of a 18 moths-old boy with SMARD phenotype. Through WES analysis they identified a new synonymous hemizygous variant in the LAS1L gene. Overall this study is very interesting, but the authors should be revise the grammar (spaces, dots and commas) and the figures (i.e. FIG.1a there is a missing part of the figure). Moreover, it's not clear why authors indicates as housekeeping gene ACTB and then in the graph (FIG.2A) is insert another gene (ATP5F1). Did the authors used more than one houskeeping gene? In that case, authors should write all the gense in the methods, and also why the normalization of actin is not included in the graph?
As mentioned above this study is very interesting and demonstrated a new mutation in LAS1L gene which could give new insight in the discovery of new pathological pathways. Nevertheless I strongly recommend a better attention to details which seems to be missing.
Author Response
Review 3
• In this article, the authors reported a clinical case of a 18 moths-old boy with SMARD phenotype. Through WES analysis they identified a new synonymous hemizygous variant in the LAS1L gene. Overall this study is very interesting, but the authors should be revise the grammar (spaces, dots and commas) and the figures (i.e. FIG.1a there is a missing part of the figure). As mentioned above this study is very interesting and demonstrated a new mutation in LAS1L gene which could give new insight in the discovery of new pathological pathways. Nevertheless I strongly recommend a better attention to details which seems to be missing.
Answer: Thank you so much for your detailed review. All comments and suggestions are extremely valuable to us. We hope that the corrections we made will be satisfying.
We check the text again and revise the grammar as well as we corrected the mistakes in the Figures.
• It's not clear why authors indicates as housekeeping gene ACTB and then in the graph (FIG.2A) is insert another gene (ATP5F1). Did the authors used more than one houskeeping gene? In that case, authors should write all the gense in the methods, and also why the normalization of actin is not included in the graph?
Answer: Two housekeeping genes were used as an endogenous control of qPCR reaction: ATP5F1 and GAPDH. The legend for Figure 2 is fine, while there is a mistake in the description of Gene expression (qPCR) section, which is now corrected as follows: “We used two housekeeping genes: ATP5F1 (Forward: 5'-GTCCAGGGGTATTACAGGCAA-3', Reverse: 5'-TCAGGAATCAGCCCAAGACG-3') and GAPDH (Forward: 5'-GCACAGTCAAGGCCGAGAAT-3', Reverse: 5' GCCTTCTCCATGGTGGTGAA-3') as endogenous control.. All 3 samples combined with 3 genes (LAS1L, ATP5F1 and GAPDH) and negative control were tested in 3 replicates.”
Reviewer 4 Report
The manuscript of “Server Infantile Axonal Neuropathy with Respiratory Failure Caused by Novel Mutation in X-linked LAS1L Gene” by Agnieszka et al. identified a hemizygous synonymous variant (c.846G>C, p. Thr282=) in LAS1L in a boy. Now limited cases presented synonymous variants that could lead to human genetic disease by forming aberrant mRNA splicing. The authors identified the novel silence variant by trio WES with the exclusion of SMARD1 and indicated it influenced the expression of LAS1L and predicted impaired RNA splicing events. It is a valuable case report for rare SMARD2 caused by LAS1L variants.
Some comments:
1. Currently, most LAS1L mutations are missense changes. How to explain the mechanism of LAS1L resulting in SMARD2? Is it dosage-sensitive for LAS1L in terms of a 50% reduction of LAS1L expression in this study?
2. In several studies of IGHMBP2, never biopsy examination showed a reduction of the myelin sheath that indicated peripheral neuropathies (Luan et al 2016, Brain and Development,38(7) 685, Pitt et al, Brain 126(12):2682). Due to limited studies on LAS1L, it is still unclear about the effect of LAS1L mutation on axonal neuropathy. Does the “Axonal Neuropathy” caused by LAS1L suggest to be changed?
3. It had better have a western blot to verify the impaired RNA splicing. Or cDNA sequencing of LAS1L from a patient.
4. How to distinguish SMARD type 2 from other similar conditions such as Infantile Respiratory Failure except SMARD1 caused by IGHMBP2? What are the other different diagnoses you have done exclusively?
5. Does any clinical report identify “seizure” of phenotype included in SMARD?
6. What are the "cytogenetic test and methylation test" at line 71 on page 3 of 8? Karyotype? What methylation loci do you consider testing?
Author Response
Review 4
• The manuscript of “Server Infantile Axonal Neuropathy with Respiratory Failure Caused by Novel Mutation in X-linked LAS1L Gene” by Agnieszka et al. identified a hemizygous synonymous variant (c.846G>C, p. Thr282=) in LAS1L in a boy. Now limited cases presented synonymous variants that could lead to human genetic disease by forming aberrant mRNA splicing. The authors identified the novel silence variant by trio WES with the exclusion of SMARD1 and
indicated it influenced the expression of LAS1L and predicted impaired RNA splicing events. It is a valuable case report for rare SMARD2 caused by LAS1L variants.
Answer: Thank you so much for your detailed review. All comments and suggestions are extremely valuable to us. We hope that the corrections we made will be satisfying.
• Currently, most LAS1L mutations are missense changes. How to explain the mechanism of LAS1L resulting in SMARD2? Is it dosage-sensitive for LAS1L in terms of a 50% reduction of LAS1L expression in this study?
Answer: According to ClinVar database (https://www.ncbi.nlm.nih.gov/clinvar) there are three variants in LAS1L classified as “likely pathogenic”, including one nonsense and two missenses. While in HGMD database (http://www.hgmd.cf.ac.uk/ac/index.php) there are four missense variants with confidence “High”, including c.1430G>A/p.S477N, described by Butterfield et al. (PMID: 24647030) as associated with SMARD-like phenotype. It is worth to emphasize, that most described cases of SMARD phenotype (MIM # 604320) are caused by homozygous or compound heterozygous mutation (including splice junction loss variants) in the IGHMBP2 gene. Since both genes, IGHMBP2 and LAS1L, play a role in ribosomal biogenesis, it is proposed that disruption of ribosomal maturation may be a common pathogenic mechanism linking
SMARD phenotypes caused by both IGHMBP2 and LAS1L. Thus, we believe that identified in our study LAS1L p.(Thr282=) variant impairing RNA splicing process may have a similar effect like splice-site mutations described in IGHMBP. Since the presence of retained intron is observed, it is expected to result
in NMD (PMID: 27450922) or, in a case of partial intron retention, a truncated protein with impaired or lack of function (PMID: 25568292). To address the Reviewer comment in the manuscript, we changed a part of Discussion section as follows: “Our case such as the case presented by Butterfield et al. suggests a common pathogenic mechanism, disruption of ribosomal biogenesis [3]. The same negative influence for ribosomal maturation have the mutations in the IGHMBP2 gene, which are responsible for most known genetic cases of SMARD
phenotype [6,10]. Thus, we expect that the identified in our study LAS1L p.(Thr282=) variant impairing RNA splicing process may have a similar effect like splice-site mutations described in IGHMBP. Since the presence of full or partial retained intron is observed, it is expected to result in NMD or truncated
protein with impaired or lack of function [11,12].”
• In several studies of IGHMBP2, never biopsy examination showed a reduction of the myelin sheath that indicated peripheral neuropathies (Luan et al 2016, Brain and Development,38(7) 685, Pitt et al, Brain 126(12):2682). Due to limited studies on LAS1L, it is still unclear about the effect of LAS1L mutation on axonal neuropathy. Does the “Axonal Neuropathy” caused by LAS1L suggest to be changed?
Answer: You have right. Thank you for this professional remark. We used one of the entities of SRAMD described on OMIM pages (severe infantile axonal neuropathy with respiratory failure) as well as we based on the main clinical symptoms of our patient such as respiratory insufficiency and severe hypotonia
resulting from neuropathy. We would like to preserve our title underlying the unknown pathomechanism of such symptoms with our molecular findings. We corrected some sentences in the main text underlying the unknown mechanism of some sypmtoms.
• It had better have a western blot to verify the impaired RNA splicing. Or cDNA sequencing of LAS1L from a patient.
Answer: Apart from qPCR, reverse transcription–polymerase chain reaction and cDNA sequencing were used to characterize the impact of the novel LAS1L variant. We performed cDNA sequencing of patient’s, patient’s mother and healthy male control using two different methods: NGS-based amplicon deep
sequencing (ADS) and Sanger direct sequencing. We improved description of methods and results section to highlight cDNA sequencing, which are now as follows:“In addition to qPCR we sequenced cDNA of patient, patient’s mother and healthy control using NGS- based amplicon deep sequencing and Sanger sequencing methods to check the absence of any coding regions. “
And “Sanger sequencing did not reveal any missing exons that could resulted from affected splicing. However, qPCR based on fibroblasts material showed a 50% decrease of LAS1L gene expression in the proband comparing to his healthy control (Figure 2A). Furthermore, results from cDNA deep amplicon sequencing identify a retained intron (between exon 6 and 7) that should result in a codon stop in the next 22 amino acids (Figure 2B).”
• How to distinguish SMARD type 2 from other similar conditions such as Infantile Respiratory Failure except SMARD1 caused by IGHMBP2? What are the other different diagnoses you have done exclusively?
Answer: The general condition of newborn was affected by severe respiratory insufficiency. The main symptoms of SRAMRD are weakness, respiratory failure, hypo- or areflexia as well as diaphragmatic paralysis. Metabolic and infectious reasons were excluded. Using widespread molecular tool such as WES, other rare cause of inborn errors of severe respiratory insufficiency and hypo- or areflexia were excluded.
• Does any clinical report identify “seizure” of phenotype included in SMARD?
Answer: No, you have wright. We corrected it.
• What are the "cytogenetic test and methylation test" at line 71 on page 3 of 8? Karyotype? What methylation loci do you consider testing?
Answer: Thank you for your this remarks. We corrected this sentence. cytogenetic test – karyotype, and methylation test – test against PWS (MS-MLPA)
Round 2
Reviewer 1 Report
The authors have addressed my concerns.
Author Response
- The authors have addressed my concerns.
Answer: Thank you very much for your reply. We are glad that the corrections we made are satisfying for you.
Reviewer 2 Report
There remain several important issues with the paper after the submitted reviews. The authors description of the phenotype remains insufficient. Without EMG and nerve conduction studies, it is difficult to justify diagnosis with a neuropathic disorder. The CSVT identified on MRI, alternating high and low tone, seizures, and other features are not typical of a severe neuropathy. The authors include a statement that the chest x-ray may indicate a paralyzed diaphragm, but they do not describe the findings on the x-ray, which could easily be included. Some additional data are presented for the sequencing fibroblast cDNA, but the figure could be much improved. The proband sequencing shows some included intronic sequence, but the exact breakpoint of the retained intron is not shown. The authors are proposing that a relatively minor decrease in LAS1L expression levels due to the transcript with the retained intronic sequence is causative of disease, but later acknowledge that there is no evidence of a dosage effect in LAS1L. With a phenotype that is not typical for a neuropathy and lack of clearly defined breakpoint for splicing changes or evidence of a dosage effect it is hard to justify this variant as pathogenic in this case.
Author Response
Thank you for your detail review. Below we answered you for yours remarks.
- The authors description of the phenotype remains insufficient. Without EMG and nerve conduction studies, it is difficult to justify diagnosis with a neuropathic disorder. The CSVT identified on MRI, alternating high and low tone, seizures, and other features are not typical of a severe neuropathy. The authors include a statement that the chest x-ray may indicate a paralyzed diaphragm, but they do not describe the findings on the x-ray, which could easily be included.
Answer:
The general condition of newborn was very severe. Many suggested examinations by consulted clinical geneticists were not accepted by intensive care physicians. We know that some additional test such as EMG and ENG would be helpful in the understanding the mechanism of symptoms. We changed the sentence with MRI description. We also completed some clinical sentences.
-
Some additional data are presented for the sequencing fibroblast cDNA, but the figure could be much improved. The proband sequencing shows some included intronic sequence, but the exact breakpoint of the retained intron is not shown.
Answer:
as suggested we re-analyzed the results of amplicon deep sequencing of patient’s cDNA to define the breakpoint of the retained intron and to improved Figure 2A. We observed a dramatic drop of coverage starting at the genomic position chrX:65528876 (c.846+336C) from 357x to a single reads at position g.chrX:65528875. This showed that the c.846G>C variant results in aberrant transcript containing a 336 bp partial intron 6 retention with retained intron breakpoint at g. chrX:65528876. New Figure 2 was provided and an appropriate text was added to the results section, which is now as follows: “Furthermore, results from cDNA deep amplicon sequencing identify in the proband, but not in proband’s mother, a retained intron 6 (between exon 6 and 7). Detail analysis revealed a dramatic drop of coverage starting at the genomic position chrX:65528876 (c.846+336C) from 357x to a single reads at position g.chrX:65528875, which in fact showed a 336 bp partial intron 6 retention with retained intron breakpoint at g. chrX:65528876. It is expected that the aberrant transcript results in a premature termination codon occurrence in the next 24 amino acids (Figure 2B).”
-
The authors are proposing that a relatively minor decrease in LAS1L expression levels due to the transcript with the retained intronic sequence is causative of disease, but later acknowledge that there is no evidence of a dosage effect in LAS1L. With a phenotype that is not typical for a neuropathy and lack of clearly defined breakpoint for splicing changes or evidence of a dosage effect it is hard to justify this variant as pathogenic in this case.
Answer:
we did not find any reported evidences of LAS1L to be sensitive for dosage effect. However, our results showed that the c.846G>C leads to 50% reduction of LAS1L expression, thus exerts effect on dosage of LAS1L mRNA. It may be concluded, that in our patient pathology of SMARD-like phenotype may contain dosage effect of LAS1L mRNA through NMD. The following text was added to the Discussion section: “We also confirmed that alteration of splicing leading to decreased dosage of LAS1L transcripts via (the most probably) NMD may be a molecular pathomechanism of SMARD-like phenotype observed in our patient.”